# Potential Driving Factors on Surface Solar Radiation Trends over China in Recent Years

**Qiuyan Wang** [1,2,3], **Hua Zhang** [1,2,*], **Su Yang** [4], **Qi Chen** [2], **Xixun Zhou** [2], **Guangyu Shi** [1,5], **Yueming Cheng** [6] and **Martin Wild** [3]

1 Collaborative Innovation Center on Forecast and Evaluation of Meteorological Disasters, Nanjing University of Information Science and Technology, Nanjing 210044, China; wqy_ncc@163.com (Q.W.); shigy@mail.iap.ac.cn (G.S.)
2 State Key Laboratory of Severe Weather, Chinese Academy of Meteorological Sciences, Beijing 100081, China; chenqi@cma.gov.cn (Q.C.); abzhouxixun@163.com (X.Z.)
3 Institute for Atmospheric and Climate Science, ETH Zurich, 8092 Zurich, Switzerland; martin.wild@env.ethz.ch
4 National Meteorological Information Center, China Meteorological Administration, Beijing 100081, China; yangsu@cma.gov.cn
5 State Key Laboratory of Numerical Modeling for Atmospheric Sciences and Geophysical Fluid Dynamics, Institute of Atmospheric Physics, Chinese Academy of Sciences, Beijing 100101, China
6 Key Laboratory of Meteorological Disaster of Ministry of Education, Nanjing University of Information Science and Technology, Nanjing 210044, China; chengyueming@mail.iap.ac.cn
* Correspondence: huazhang@cma.gov.cn

**Abstract:** The annual mean surface solar radiation (SSR) trends under all-sky, clear-sky, all-sky-no-aerosol, and clear-sky-no-aerosol conditions as well as their possible causes are analyzed during 2005–2018 across China based on different satellite-retrieved datasets to determine the major drivers of the trends. The results confirm clouds and aerosols as the major contributors to such all-sky SSR trends over China but play differing roles over sub-regions. Aerosol variations during this period result in a widespread brightening, while cloud effects show opposite trends from south to north. Moreover, aerosols contribute more to the increasing all-sky SSR trends over northern China, while clouds dominate the SSR decline over southern China. A radiative transfer model is used to explore the relative contributions of cloud cover from different cloud types to the all-types-of-cloud-cover-induced (ACC-induced) SSR trends during this period in four typical sub-regions over China. The simulations point out that the decreases in low-cloud-cover (LCC) over the North China Plain are the largest positive contributor of all cloud types to the marked annual and seasonal ACC-induced SSR increases, and the positive contributions from both high-cloud-cover (HCC) and LCC declines in summer and winter greatly contribute to the ACC-induced SSR increases over East China. The contributions from medium-low-cloud-cover (mid-LCC) and LCC variations dominate the ACC-caused SSR trends over southwestern and South China all year round, except for the larger HCC contribution in summer.

**Keywords:** SSR trends under different conditions; cloud and aerosols; radiative transfer model; relative contributions; different types of cloud cover

## 1. Introduction

As an important component of the climate system, solar radiation incident at the Earth's surface is the primary energy source for life on the planet. It can influence surface temperature, the hydrological cycle, plant photosynthesis and carbon uptake, thereby largely determining our climatic conditions and ecological environment [1–3]. However, this quantity is not constant at decadal time scales, instead, it has undergone significant decadal variations since the 1950s in many regions of the world, declining until the late

1980s and recovering thereafter, popularly known as "global dimming" and "brightening" [2–5].

Many efforts have been made to explore the reasons for such changes in surface solar radiation (SSR) over the past decades. Studies show that little changes occur in solar radiation arriving at the top of the atmosphere [6], whereas SSR has experienced dimming and brightening over time [2]. Changes in the transparency of the atmosphere are therefore the major contributors to the SSR variations over decades, including changes in cloud characteristics (e.g., cloud cover and cloud optical properties), mass concentration and optical properties of aerosols, and radiatively active gases in the atmosphere [3]. However, the relative importance of the above-mentioned factors mainly depends on the meteorological conditions or the degree of air pollutions [3,5].

East Asia, especially China, is not only a high-emission region of greenhouse gases and air pollutants but also deeply affected by the monsoon climate. The measured SSR trends over China have been examined by many previous studies, characterized by significant dimming between the 1950s and late 1980s with subsequent brightening [7–10]. However, many recent studies reported that the ground-based SSR observations in China have suffered from substantial inhomogeneity issues mainly induced by instrument replacement and instrument sensitivity drift especially during 1990–1993, and many efforts have been made to diminish the inhomogeneities [11–17]. As a result, all these recent studies suggested that SSR in China decreased markedly over 1961–1990, and remained stable afterward or continued to decrease until the 2000s with a brightening in recent years.

Many studies arrived at the consistent conclusion that an increase in aerosol loading in the atmosphere is a major driving factor for the dimming period in China [7,8,10,18,19]. However, China has implemented many measures to control aerosol emissions in recent years. Some studies illustrated the importance of clouds on the brightening phase in China [9,20–23]. Norris and Wild [20] inferred that half of the brightening trends for 1990–2002 over China could be attributed to a reduction in cloud cover. Xia [21] examined the long-term changes in cloud amount and sunshine duration (SSD) over China, indicating that low cloud cover appeared to be one of the major causes of SSD trends in southern China. Wang et al. [22] suggested that SSR in most regions of China began to increase after 1990, and reductions in cirrus and cirrostratus had distinct contributions to the trends over 1990–2000. Yang et al. [9] concluded that clouds play a more important role in the brightening phase over 1990–2009 in China. Zhang et al. [23] found that clouds contributed more to the long-term SSR changes from different satellite-derived products than aerosols. Nevertheless, by analyzing the SSR trends, as well as aerosol optical depth (AOD) and cloud amount in East China over the most recent decade from 2005 to 2015, Li et al. [19] found that a reduction in AOD was the major contributor to the brightening.

More common studies tend to suggest that the brightening trends in SSR over China are mainly due to the combined effects of clouds and aerosols. Wang et al. [24] pointed out that the brightening since the 1990s was associated with cloud suppression and reduction in anthropogenic emissions. Tang et al. [25] found that changes in cloud properties and interactions between clouds and aerosols rather than aerosols or water vapor were likely to be the primary causes of the SSR variations over the period 1980–2010 in China. Yang et al. [17] indicated that the clouds counteracted aerosol effects since 2000, based on homogenized daily SSR data under clear- and all-sky conditions. Therefore, the relative importance of aerosols and clouds on the brightening period over China still needs further investigation, primarily depending on different data sources or data quality assurances.

The SSR trend analyses above are mainly based on surface observations from the China Meteorological Administration (CMA). The temporal and spatial coverage of SSR has greatly improved thanks to the advent of the satellite era since the early 1980s. Clouds and Earth's Radiant Energy System (CERES) products are considered to show the best agreement with observed and SSD-derived SSR among the model-based/satellite-derived SSR estimates due to their better performance in cloud parameters [13,23], followed by the reanalyses, particularly Modern-Era Retrospective analysis for Research and Applications,

Version 2 (MERRA2) [26], and the Coupled Model Intercomparison Project Phase 5 (CMIP5) multi-model estimates [13]. Furthermore, Yang et al. [16] reported that the transition year from dimming to brightening over China occurred in 2005 by using monthly homogenized SSR data, which provides the exact starting year of the brightening period in China for this study. Besides, a transition to decrease in the PM2.5 concentrations has been reported in China after 2005 [27]. Previous studies never quantitatively provided the contributions of different driving factors to the SSR variations in different regions of China. Therefore, different satellite-derived products are used to identify the possible causes (e.g., cloud cover, AOD, water vapor, and ozone($O_3$)) of the brightening period from 2005 to 2018 over China. Then, the annual and seasonal means of these factors for each year during this period are used as inputs into a radiative transfer model to calculate their respective effects on SSR—especially since only a few studies ever explored the effects of cloud cover from different cloud types on SSR variations quantitively [21,22]. This study thereby examines the relative contributions of cloud cover from different cloud types to the sum of absolute trends induced by each cloud cover type (thereafter using "all-types-of-cloud-cover-induced (ACC-induced) SSR trends" instead) during the period 2005–2018 over China, in particular over four sub-regions, based on model calculations.

The paper is organized as follows: the model description, data, and methodology are given in Section 2. The validation of CERES-derived all-sky SSR with the ground-based measurement from the CMA, the determination of the major driving factors based on the CERES-derived SSR trends under different conditions and satellite-derived changes in the factors affecting them, a brief comparison of the model-based and satellite-derived relative SSR trend percentages due to different factors, and the simulated relative contributions of cloud cover from different cloud types to the ACC-induced SSR trends are presented in Section 3. Finally, conclusions are shown in Section 4.

## 2. Materials and Methods

### 2.1. Description of Radiative Transfer Model

The radiative transfer model used in this study is BCC_RAD (Beijing Climate Center radiative transfer model) [28–30]. It divides the 10–50,000 $cm^{-1}$ wavelength range into 17 wavebands, of which bands 1–8 are longwave and the others are shortwave bands. Five major greenhouse gases, i.e., water vapor, carbon dioxide ($CO_2$), $O_3$, nitrous oxide ($N_2O$), and methane ($CH_4$), and four chlorofluorocarbon (CFCs) gases as well as carbon monoxide (CO) and oxygen ($O_2$) are included in the model by considering their line absorption and continuum absorption. Furthermore, the gas absorption and overlap schemes were calculated using a correlated K-distribution method [28,31]. The optical properties of water and ice clouds were given by Lu et al. [32] and Zhang et al. [33]. The aerosol radiative scheme was adopted from Wei and Zhang [34], Zhang et al. [35], and Zhou et al. [36]. The cloud vertical overlap was dealt with a semi-random method from Nakajima et al. [37]. The radiative transfer algorithm for this study was also from Nakajima et al. [37]. For the detail of the model introduction, please see Zhang [38].

The BCC_RAD was used to investigate radiative forcings due to aerosols [39–45], the radiative effects due to changes in water and ice cloud optical thickness in East Asia [46], and the short-term cloud feedback in East Asia by using cloud radiative kernels [47]. This indicates that BCC_RAD has the ability to adequately simulate aerosol and cloud radiative effects in this study.

### 2.2. Datasets

#### 2.2.1. Reference SSR Datasets

The monthly ground-based SSR data for 99 sites across China was available from the CMA with rigid data quality control, including the spike value test, the stuck value test, and the spatial consistency test, as well as homogenization using quantile-matching adjustment (details in [16,17]).

The monthly satellite-derived SSR for all-sky, clear-sky, all-sky-no-aerosol, and pristine (clear-sky-no-aerosol) conditions during 2005–2018 in China was obtained from the CERES Edition4.1 SYN1deg monthly product at $1° \times 1°$ resolution. It is noted that these surface radiative fluxes are retrieved from the NASA Langley Fu-Liou radiative transfer model based on inputs from Moderate Resolution Imaging Spectroradiometer (MODIS) and Goddard Earth Observing System (GEOS) cloud properties, GEOS atmosphere and skin temperature, Model of Atmospheric Transport and Chemistry (MATCH) aerosol constituents, and MODIS spectral aerosol optical depths. In addition, the bias of the computed monthly mean downward fluxes from SYN1deg is 3.0 W m$^{-2}$ (5.7%) for shortwave and $-4.0$ W m$^{-2}$ (2.9%) for longwave compared to surface observations [48].

### 2.2.2. Input Datasets to the Radiative Transfer Model

The inputs of atmospheric profiles for the BCC_RAD radiative transfer model mainly include the number of vertical layers, solar zenith angle, as well as height, pressure, temperature, particle parameters (concentrations or effective radius for different particles), cloud cover, and gas concentrations for each layer, and additional surface level for surface pressure, surface temperature, skin temperature, and surface albedo. Other parameters, like calculation parameters, band divisions, spectral weight, gas absorption, and aerosol optical properties, are contained in a prescribed input parameter file for fast model calculations. Therefore, variations in the input parameters of the atmospheric profiles will satisfy the purpose of this study.

To obtain more realistic atmospheric profiles for radiative transfer calculations, as can be seen from Table 1, the input parameters in this study during the period 2005–2018 over China are taken from various datasets which dealt with the same resolution of $1° \times 1°$, including space-based observations and reanalyses. For instance, the total (TCC), high (HCC) (at levels of 50–300 mb), medium-high (mid-HCC) (300–500 mb), medium-low (mid-LCC) (500–700 mb), and low cloud cover (LCC) (700-surface mb) are from the CERES SYN1deg monthly product. The AOD is from MODIS/Aqua MYD08_M3 data. The water vapor, $O_3$, and temperature (T) data are taken from the Atmospheric Infrared Sounder (AIRS) version 6 level 3 standard product and the MERRA-2 instM_3d_ana_Np monthly mean reanalysis, while $CH_4$ and CO are only from the AIRS products. Especially, when using the AIRS data as inputs to the model, the upper atmosphere above 100 (1) mb for water vapor ($O_3$ and T) is filled with MERRA-2 data due to the lack of retrievals at these pressure levels [49]. The surface albedo, surface pressure, 2-m temperature, and skin temperature are from the European Centre for Medium-Range Weather Forecasts (ECMWF) Interim Re-Analysis (ERA-Interim). The solar zenith angle and liquid water/ice cloud effective radius, as well as cloud liquid/ice water content, are obtained from MODIS/Aqua and CloudSat 2B-CWC-RO data, respectively. Moreover, concentrations of other gases, namely $CO_2$, $N_2O$, and $O_2$ are set to 391, 0.324, and $0.209 \times 10^6$ ppmv, respectively, mainly according to the Intergovernmental Panel on Climate Change Fifth Assessment Report (IPCC AR5) [50].

These parameters above are interpolated to 66 fixed pressure levels in the vertical direction with vertical resolutions of 0.25 km at the heights below 2 km and of 1 km at the heights between 2 and 60 km. A surface level is also added to the atmospheric profiles.

Additionally, all the datasets presented in this paper have widely been assessed in previous studies. For example, Sayer et al. [51] found the MODIS AOD bias was less than 0.01 in East Asia compared to the AERONET observation network. Zhao and Zhou [52] indicated that MERRA2 performs well in reproducing the annual climatology of the total column water vapor (TCWV) (R = 0.99). Wargan et al. [53] pointed out that the $O_3$ reanalysis from MERRA2 shows agreement within 10% with independent satellite data in most of the stratosphere. Other data sources, including the ERA-Interim, Cloudsat, and AIRS L3, also have been evaluated in different ways [54,55]. Thus, the annual or seasonal variations of these datasets remain reliable although some certain deviations exist in them.

**Table 1.** Summary of input parameters for atmospheric profiles used in the BCC_RAD (Beijing Climate Center radiative transfer model).

| Variables | Data Sources | Spatial-Temporal Resolution | Dimensions | Level Ranges |
|---|---|---|---|---|
| Water vapor, $O_3$ T CO, CH4 | AIRS L3 | $1° \times 1°$/monthly | lev, lat, lon | 1000–100 hPa 1000–1 hPa 1000–0.1 hPa |
| Water vapor $O_3$, T | MERRA2 | $1° \times 1°$/monthly | lev, lat, lon | 100–0.1 hPa 1–0.1 hPa |
| Surface albedo Surface pressure 2-m temperature Skin-temperature | ERA-Interim | $1° \times 1°$/monthly | lat, lon | None |
| Solar Zenith angle Liquid water/Ice cloud effective radius Aerosol optical depth at 550 nm | MODIS/Aqua | $1° \times 1°$/monthly | lat, lon | None |
| Liquid water/Ice content | CloudSat | $2.8° \times 2.8°$/daily | lat, lon | None |
| Total/High/Mid-High/Mid-Low/ Low cloud cover | CERES SYN1deg | $1° \times 1°$/monthly | lat, lon | None |

### 2.2.3. Methodology

When calculating the cloud cover effects on changes in SSR from 2005 to 2018 in China using BCC_RAD, we only change the parameters of annual and seasonal mean cloud cover from different cloud types for each year with all other parameters in atmospheric profiles fixed at their multi-year/seasonal means in the model (without inputs of aerosol-related parameters here), respectively. Similarly, when studying AOD, water vapor, and $O_3$ impacts on SSR changes, only the annual means of these parameters vary yearly (without inputs of cloud-related parameters). Besides, the vertical heights of HCC, mid-HCC, mid-LCC, and LCC are set at 8–11 km, 6–7 km, 4–5 km, and 1–3 km in the model based on [56], respectively, while it is mainly in the troposphere for that of aerosols, especially with a vertical resolution of 0.25 km at the heights below 2 km. However, the MODIS AOD data cannot be used directly as an input to the model since it is an integrated variable over the vertical atmospheric column. Due to the lack of observations on the vertical structure, the AOD vertical-weighted profiles for various aerosol species are referred to the AOD vertical distributions simulated for the year 2006 from the Non-hydrostatic Icosahedral Atmospheric Model (NICAM) coupled with the Spectral Radiation Transport Model for Aerosol Species (SPRINTARS) [57]. Thus, it should be noted here that the AOD vertical-weighted profiles of various aerosol species in this study are not changed annually.

To calculate the effects of long-term changes in different potential driving factors (e.g., cloud cover, AOD, water vapor, and $O_3$) on SSR trends quantitatively during 2005–2018 in China, the yearly annual or seasonal means of these factors, along with their counterparts of multi-year/seasonal averages of background atmospheric profiles, were used as inputs into the BCC_RAD radiative transfer model, respectively. In this study, the concept of relative trend percentage (as defined in Formula (1)) was utilized to avoid the differences in absolute values obtained from various model assumptions.

$$Relative\ Trend\ percentage_k = \frac{Trend_k}{\sum abs(Trend_i)} \times 100\% \qquad (1)$$

where the subscript $i$ ($k$) denotes one of the SSR trends mentioned below, and the denominator represents the summation of absolute values of either the simulated SSR trends caused by the changes in TCC, AOD, water vapor, and $O_3$, or the CERES-derived SSR trends under all-sky-no-aerosol, clear-sky, and clear-sky-no-aerosol conditions or the SSR trends induced

by the HCC, mid-HCC, mid-LCC, and LCC changes for simulating contributions of cloud cover from different cloud types, respectively.

All trends in this study are linearly fitted by the least square method according to the function $y(x) = a + b \cdot x$, and b is the trend term:

$$b = \frac{\sum_{i=1}^{n}(x_i - \overline{x})(y_i - \overline{y})}{\sum_{i=1}^{n}(x_i - \overline{x})^2} \tag{2}$$

where $n$ denotes the duration of the sequence, $x_i$ ($y_i$) and $\overline{x}$ ($\overline{y}$) refer to the $i$th and average of variable $x$ ($y$), respectively. The positive/negative value of b indicates an increasing/decreasing trend, and the absolute value of b represents the magnitude of the trend. Moreover, a $t$-test of the regression coefficients at the 5% or 10% significance level is used in this study.

The simulations by radiative transfer model are thereby mainly focused on the relative SSR trend contributions of cloud cover from different cloud types to the ACC-induced SSR variations owing to the lack of yearly varying AOD vertical-weighted profiles of various aerosol species. Furthermore, four typical sub-regions were selected for the sake of more detailed analyses.

China in this work is defined as the region 17°N–55°N, 72°E–136°E. Spring, summer, autumn, and winter in this study represent March-May (MAM), June-August (JJA), September-November (SON), and December-February (DJF), respectively.

Please note the naming rules of different regions over China in this paper, in particular, the regions with longitudes greater than around 97°E are regarded as the eastern half of China, while the rest are the western half of China. Other regions, such as eastern, northeastern, southern, southwestern China, etc., are conventional and known to all.

## 3. Results

### 3.1. Validation of the CERES Dataset Using CMA Observations

A homogenized ground-based monthly SSR dataset covering 99 stations across China from the CMA during the period 2005–2018 is utilized to evaluate the performance of the CERES-derived all-sky SSR. Figure 1 shows the spatial distributions of the annual mean all-sky SSR and its corresponding trends during this period over China from the CMA (black circles) and CERES SYN1deg (contour plots) datasets, respectively. Moreover, the corresponding histograms along with the time series of area-weighted average SSR anomalies are further provided in Figure 2.

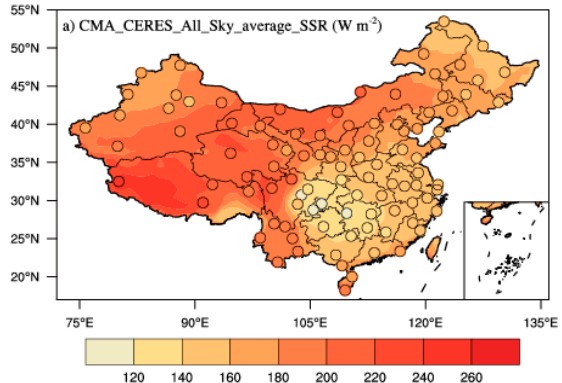 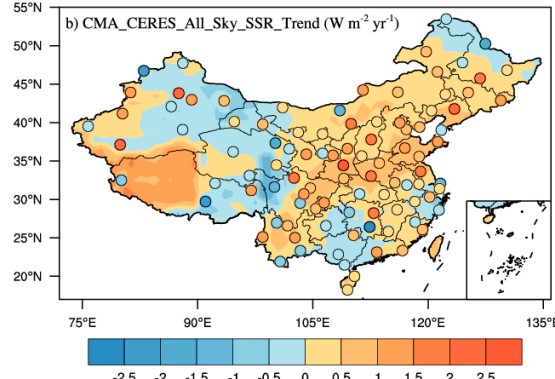

**Figure 1.** Annual mean (**a**) surface solar radiation (SSR) (unit: W m$^{-2}$) and (**b**) its corresponding trends (unit: W m$^{-2}$ yr$^{-1}$) under all-sky conditions for the period 2005–2018 over China. The black circles and contour plots represent the China Meteorological Administration (CMA) ground-based observations and the Clouds and Earth's Radiant Energy System (CERES) satellite-derived product, respectively.

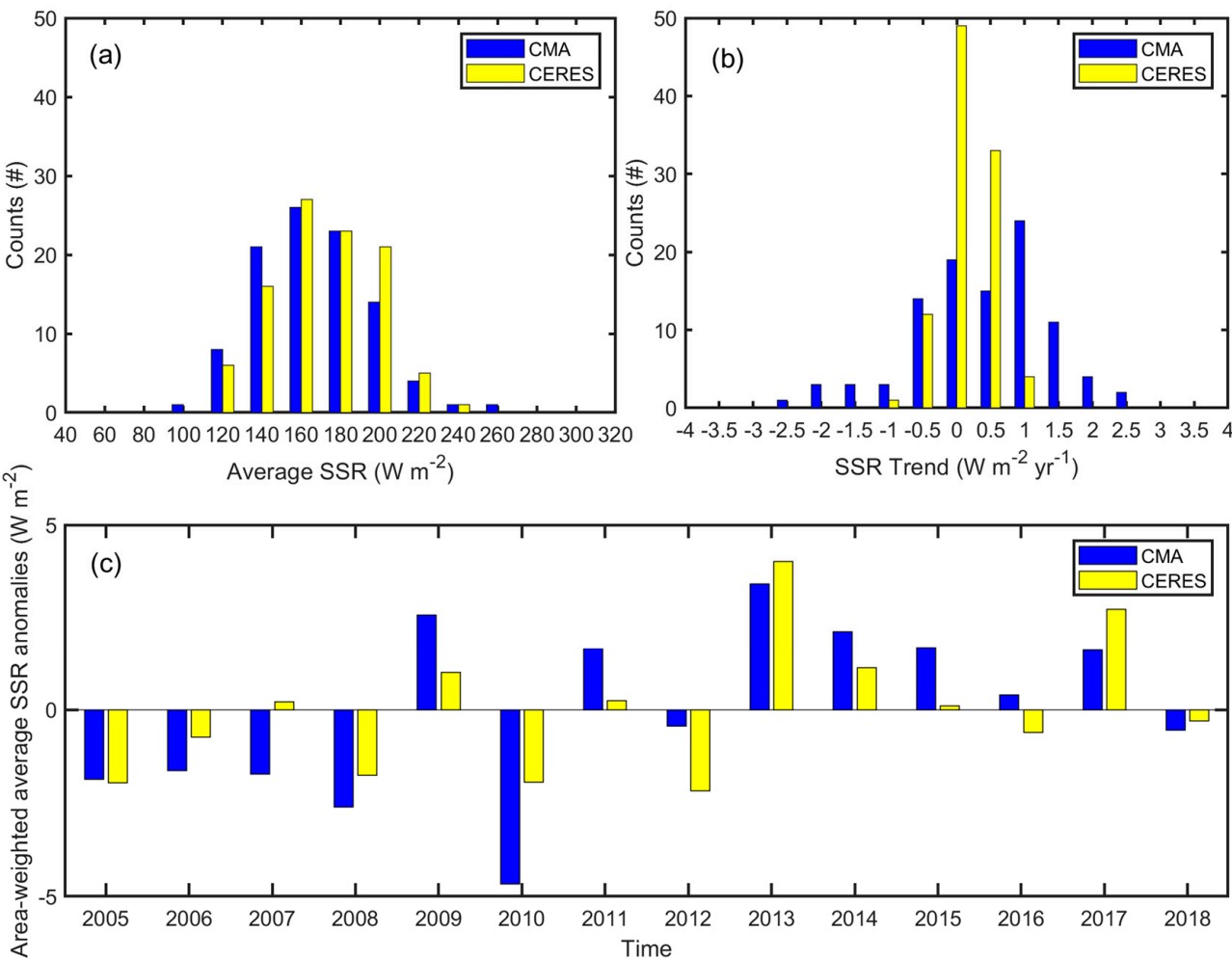

**Figure 2.** Annual mean histograms of (**a**) average SSR (unit: W m$^{-2}$) and (**b**) its corresponding trends (unit: W m$^{-2}$ yr$^{-1}$) (counts on the vertical axis represent numbers of surface sites (a total of 99 sites)) as well as (**c**) time series of area-weighted average SSR anomalies (unit: W m$^{-2}$) under all-sky conditions during the period 2005–2018 over China from the CMA (blue color) and CERES SYN1deg product (yellow color), respectively.

As shown in Figures 1a and 2a, the spatial and frequency distributions of annual mean all-sky SSR from the CERES product shift toward higher values compared to that of the CMA stations, which agrees well with the higher annual area-weighted average SSR (175 VS 170 W m$^{-2}$) (Table 2). This rightward shift can also be found in the histograms of spring, autumn, and winter mean all-sky SSR but with a similar frequency distribution in summer (Figure S3), thus contributing to much closer area-weighted all-sky SSR values in the summer and autumn seasons (Table 2). In addition, the CERES-estimated SSR variations are mainly within ±0.5 W m$^{-2}$ yr$^{-1}$ with a rightward shift, while a similar shift appears with a border distribution in the CMA trends (Figure 2b), resulting in a smaller area-weighted trend in Table 2 (0.27 VS 0.18 W m$^{-2}$ yr$^{-1}$). Similar explanations also apply to the seasonal patterns of the histogram (Figure S3 and Table 2).

**Table 2.** The area-weighted average SSR (including the standard deviation, Units: W m$^{-2}$) and its corresponding trends (Units: W m$^{-2}$ yr$^{-1}$) under all-sky conditions over China during the period 2005–2018 at annual and seasonal time scales from the CMA and CERES SYN1deg data, respectively. The CMA station data is first interpolated onto a 1° × 1° grid, and then the area-weighted averages are calculated based on the interpolation data.

| During the Period 2005–2018 over China | CMA | | CERES SYN1deg | |
|---|---|---|---|---|
| | Average SSR (W m$^{-2}$) | SSR Trends (W m$^{-2}$ yr$^{-1}$) | Average SSR (W m$^{-2}$) | SSR Trends (W m$^{-2}$ yr$^{-1}$) |
| ANN | 170 ± 2.31 | 0.27 | 175 ± 1.80 | 0.17 |
| MAM | 202 ± 4.01 | 0.35 | 209 ± 3.15 | 0.33 |
| JJA | 222 ± 3.82 | 0.34 | 223 ± 3.63 | 0.33 |
| SON | 147 ± 2.06 | −0.06 | 151 ± 1.93 | −0.2 |
| DJF | 108 ± 2.96 | 0.45 | 116 ± 2.96 | 0.2 |

According to the spatial distributions of the annual mean all-sky SSR trends in Figure 1b, the CERES-derived dataset generally shows a good performance over the eastern half of China (the definition of this region can be seen in the last paragraph of Section 2) compared to the ground-based observations from the CMA, except for some western regions of China as well as individual regions over central and eastern China. The performances of the spring and summer mean CERES-derived SSR trends under all-sky conditions are better than those in autumn and winter seasons, which is possibly induced by the opposite trends over western and northeastern China (Figure S2). The biases over northeastern China might be related to the fewer sampling and higher retrieval uncertainty of surface albedo induced by longtime coverage of snow in the cold seasons [19]. The reasons for discrepancies over northwestern and southwestern China are likely due to difficulty in evaluating AOD on the variable and high-albedo surface as well as the incapable consideration of elevation impacts on the satellite algorithms and degraded data quality under snow-cover surfaces [58]. For eastern China, the difference may result from an improper representation of AOD induced by rapid economic growth in this region [58,59]. Furthermore, the time series of anomaly area-weighted average SSR for the studying period is further investigated to explore the exact variation for each year. Figure 2c and Figure S4 indicate that the annual and seasonal mean area-weighted average SSR from both the CMA and satellite datasets over China varies simultaneously with different magnitudes, apart from one or two certain years. However, this only represents the national average condition, it would be another case for some specific regions.

In short, the performance of average SSR under all-sky conditions is better than that of its collocated trends during the period 2005–2018 over China. However, they correspond reasonably well to the ground-based observations both at the annual and seasonal time scales over the eastern half of China, although some discrepancies exist in some individual regions. Therefore, the following analyses of CERES product would focus on the eastern half of China for better accuracies.

*3.2. Analysis of Satellite-Derived SSR Trends under Different Conditions and Their Potential Causes*

To analyze the potential causes of such SSR variations over these regions in recent years, the spatial patterns of annual SSR trends under different conditions derived from CERES-derived product (Figure 3) and corresponding changes in TCC, AOD, water vapor, and O$_3$ from different satellite-derived products (Figure 4) are investigated. These conditions mainly represent the effects of all factors as well as the individual factors of aerosols, clouds, and gases on SSR trends. As can be seen from Figure 3a, the annual mean all-sky SSR increases during 2005–2018 over most regions in China, which is generally consistent with the surface brightening documented by previous studies based on surface radiation observations [16,17,19]. The largest significant increases occur in latitudes of approximately 40°N–43°N and 33°N–38°N over the eastern half of China with average increases around 0.6 W m$^{-2}$ yr$^{-1}$ (not shown), which is close to the national average of 0.613 W m$^{-2}$ yr$^{-1}$

in China during 2005–2016 reported by Yang et al. [16] based on station observations from the CMA. However, slight declines appear over the western Sichuan, Guangdong, Guangxi, Guizhou, and Zhejiang provinces, with the maximum decreases being up to $-0.89$ W m$^{-2}$ yr$^{-1}$ (not shown) over the western Sichuan province (see supplementary Figure S5 for the map of Chinese provinces).

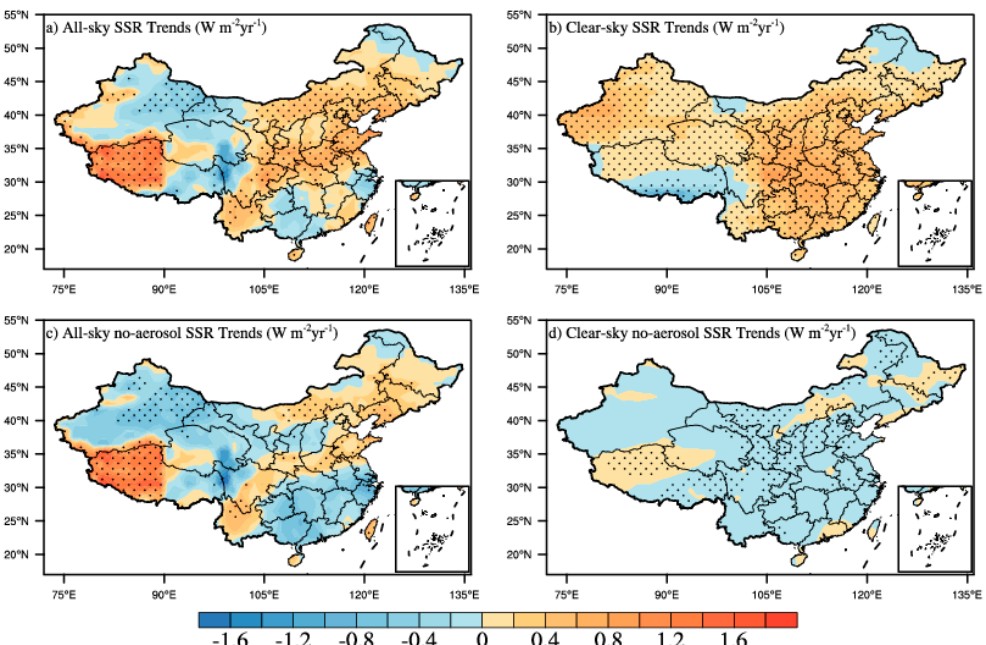

**Figure 3.** Annual mean SSR trends under (**a**) all-sky, (**b**) clear-sky, (**c**) all-sky-no-aerosol, and (**d**) clear-sky-no-aerosol conditions from 2005 to 2018 over China based on CERES SYN1deg product (unit: W m$^{-2}$ yr$^{-1}$). The dots represent that the trend is above 90% significance level from the *t*-test.

Significant widespread increases in clear-sky SSR appear over the vast majority of China with an average rate of about 0.55 W m$^{-2}$ yr$^{-1}$ (not shown), especially over central China (Figure 4b), whereas it is about half of the national average of 1.06 W m$^{-2}$ yr$^{-1}$ for 2008–2016 given by Yang et al. [17]. The difference in the magnitudes is probably caused by the biases existing in ground-based and satellite-derived datasets as well as the different methods to calculate the SSR trends. This brightening is primarily due to significant declines in AOD during this period resulting from the implementation of a series of air pollution mitigation measures in China in recent years (Figure 4b) [60,61]. Correspondingly, the satellite-derived AOD reductions are mainly distributed in eastern Sichuan and central China with decreases generally greater than 0.024 yr$^{-1}$ (not shown), which agrees well with the above clear-sky SSR increases (Figure 3b).

The regional distributions of the satellite-derived cloud effects on SSR trends seem to be more uneven compared to the aerosol effects (Figure 3b,c). The annual mean SSR due to cloud impacts is decreased by an average rate around 0.4 W m$^{-2}$ yr$^{-1}$ (not shown) over most regions of southern China, northern Shaanxi, and Shanxi provinces, whereas it is increased with a rate slightly larger than 0.2 W m$^{-2}$ yr$^{-1}$ (not shown) in most regions over southwestern China, latitudes of about 32°N–37°N, and northeastern China (Figure 3c). Accordingly, the spatial pattern of TCC trends from the CERES SYN1deg product (Figure 3a) varies from south to north, characterized by marked increases over southern China, in particular over the Guangdong and Guangxi provinces with the maximum reaching 0.008 yr$^{-1}$ (not shown), as well as decreases over most regions of northern and northeastern China with a decline generally around $-0.004$ yr$^{-1}$ (not shown), which is largely in line with SSR changes caused by clouds (Figure 3c) due to both their absorbing and reflecting properties.

Thus, the SSR variations induced by clouds (Figure 3c) can be generally explained by the changes in TCC over these regions during this period in China (Figure 4a).

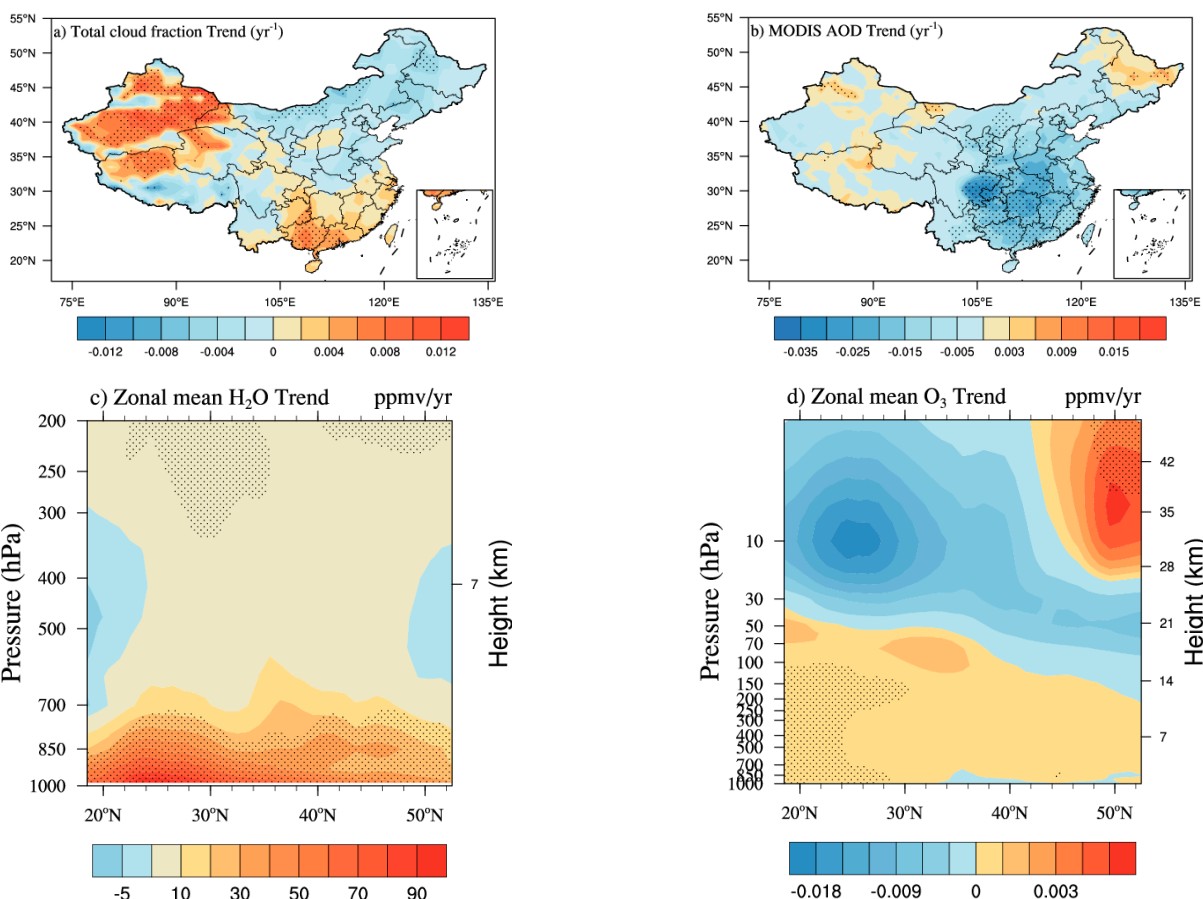

**Figure 4.** Annual mean trends of (**a**) total cloud cover (TCC) (unit: $yr^{-1}$) from CERES, (**b**) aerosol optical depth (AOD) (unit: $yr^{-1}$) from the Moderate Resolution Imaging Spectroradiometer (MODIS), zonally averaged (**c**) water vapor (unit: ppmv $yr^{-1}$), and (**d**) ozone ($O_3$) (unit: ppmv $yr^{-1}$) from a combination of Atmospheric Infrared Sounder (AIRS) and Modern-Era Retrospective analysis for Research and Applications, Version 2 (MERRA2) products for 2005–2018 over China. The dots represent that the trend is above 95% significance level from the *t*-test.

The magnitudes of SSR changes induced by radiatively active gases are very small, with an average declining rate of $-0.03$ W m$^{-2}$ yr$^{-1}$ over China (Table 3), compared to the SSR trends caused by aerosols and clouds (Figure 3d,b,c). The changes by water vapor and $O_3$ are the major gases that could make a small contribution to the satellite-derived SSR variations according to previous sensitivity studies with radiative transfer models [3,17,62,63]. Water vapor and $O_3$ play important roles in the process of shortwave radiation transmission due to the absorptive effects of water vapor in the near-infrared and $O_3$ in the visible and ultraviolet wavelength ranges. As shown in Figure 4c, the zonal mean water vapor increases substantially in the atmospheric levels lower than 700 hPa and decreases with altitudes during this period, with the largest increases exceeding 90 ppmv yr$^{-1}$ (not shown). The changes in the vertical distribution of $O_3$ vary among pressure levels, featured by increases in the troposphere and lower stratosphere as well as declines in the higher stratosphere, with the maximum and minimum rates up to 0.001 and $-0.018$ ppmv yr$^{-1}$, respectively (Figure 4d). Therefore, the changes in water vapor and $O_3$ will lead to a small reduction and increase in SSR during this period over most regions in China, respectively. Combined with the satellite-derived declining SSR trends due to the

gases (Figure 3d), we can infer that the increase in water vapor contributes more to such SSR trends induced by gases than that of $O_3$.

**Table 3.** The annual mean median and average SSR trends (Units: W m$^{-2}$ yr$^{-1}$) over China under different conditions during the period 2005–2018 from CERES SYN1deg product.

| Annual SSR Trends over China for 2005–2018 (W m$^{-2}$ yr$^{-1}$) | Median Trends | Average Trends |
|---|---|---|
| under All-sky condition | 0.13 | 0.18 |
| under Clear-sky condition | 0.19 | 0.23 |
| under All-sky-no-aerosol condition | −0.02 | −0.02 |
| under Clear-sky-no-aerosol condition | −0.04 | −0.03 |

As can be seen from Table 3, both the annual mean median and average SSR trends over China for the period 2005–2018 under all-sky condition are approximately the sum of those under clear-sky, all-sky-no-aerosol, and clear-sky-no-aerosol conditions (0.13, 0.19, −0.02, and −0.04 W m$^{-2}$ yr$^{-1}$ as well with 0.18, 0.23, −0.02, and −0.03 W m$^{-2}$ yr$^{-1}$, respectively), indicating that the annual mean nationwide brightening in China during this period is mainly due to reductions in aerosols, while clouds also contribute substantially to the brightening over most regions of eastern and northeastern China but with opposite contributions over southern China. This is also the reason why the magnitude of the national mean contributions from clouds is very small and close to that of radiatively active gases.

In general, aerosols and clouds may be of different importance for SSR variations in different regions of China. According to the trend analysis above, changes in clouds and aerosols are responsible for the marked increases in all-sky SSR over most regions of northern and northeastern China, and aerosols seem to play a more important role in such SSR trends on average. However, cloud variations play an opposite role in SSR trends over southern China, northern Shanxi, and Shanxi provinces. Moreover, this study indicates that clouds and aerosols are the major contributors to the SSR trends, whereas water vapor and $O_3$ play an insignificant role. The same conclusions were also drawn by Yang et al. [17], namely that water vapor cannot be the major cause of the long-term dimming and brightening in China.

*3.3. Comparisons of the Model-Estimated and Satellite-Derived Relative SSR Trend Percentages Due to Different Factors*

In order to examine the performance of the BCC_RAD radiative transfer model in its abilities to simulate SSR trends due to different driving factors, the model outputs are compared with reference values from CERES products in terms of relative SSR trend percentages to prevent discrepancies in absolute values induced by different assumptions. Besides, the simulations at high-altitude regions in China are not as accurate as expected due to the shortage of the vertical inputs to the model, in particular inputs of LCC data over the Tibetan Plateau (TP). Thus, the TP area is masked in the following figures and the focus of the analysis is mainly over the eastern half of China.

Figure 5 displays the annual mean SSR trend percentages caused by TCC, AOD, water vapor, and $O_3$ relative to the sum of absolute trends due to each factor above (thereafter using "total SSR trends" instead) over the period 2005–2018 in China as simulated by the BCC_RAD radiative transfer model, as well as the approximately corresponding relative SSR trend percentages due to clouds and aerosols derived from the CERES SYN1deg product. As can be concluded from Figure 5a–d, the simulated spatial patterns of annual mean relative SSR trend percentages due to these factors are opposite to their respective trends as shown in Figure 4. However, some differences exist in individual regions, especially over southeastern China, when comparing simulated SSR trend percentages due to TCC with TCC trends (Figures 4a and 5a). This is because the simulated TCC effects on SSR is calculated from inputs of cloud covers from all cloud types rather than only inputs

of TCC. Both the simulated relative SSR trend contributions from TCC and AOD account for large proportions of the total SSR trends over this period in the eastern half of China, while the effects of water vapor and $O_3$ make up much smaller contributions (Figure 5a–d). This also coincides with the results derived from satellite products in the above section and previous sensitivity modeling studies.

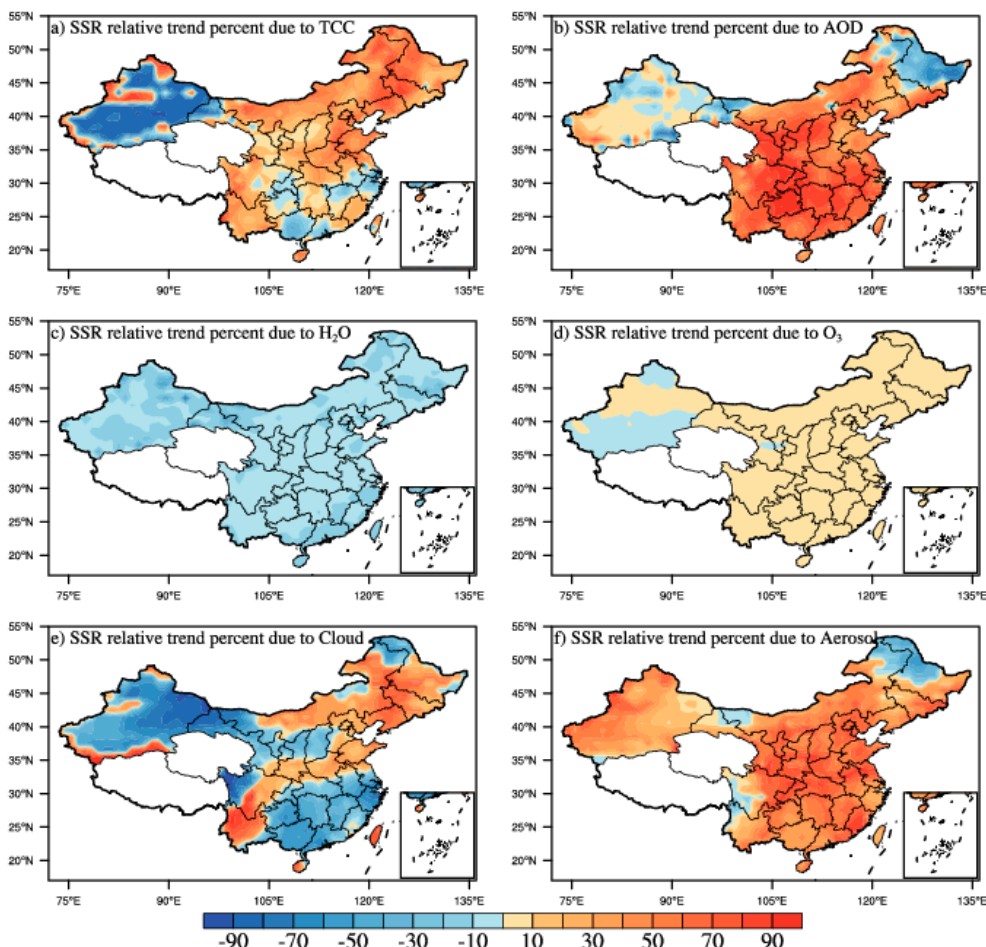

**Figure 5.** Annual mean simulated SSR trend percentages (unit: %) due to (**a**) TCC, (**b**) AOD, (**c**) water vapor, and (**d**) $O_3$ relative to the sum of absolute trends due to each of these factors (total SSR trends) from a single column radiative transfer model, as well as SSR trend percentages (unit: %) from (**e**) clouds and (**f**) aerosols relative to the sum of absolute trends due to clouds, aerosols, and gases derived from CERES SYN1deg product during 2005–2018 over China. The blue labeled sub-regions represent the (1) northeastern China, (2) central China, (3) East China, and (4) South China, which occupy latitudinal ranges of 41–45°N, 32–35°N, 27.5–32.5°N, and 22.5–25.5°N, and longitudinal ranges of 115–123°E, 105–115°E, 116–121°E, and 107–115°E, respectively.

In addition, the simulated relative SSR trend contributions from TCC and AOD to the total SSR trends (Figure 5a,b) are compared with their approximately corresponding contributions of clouds and aerosols derived from CERES (Figure 5e,f). One more thing that should be noted here is that the effects of clouds and aerosols on SSR trends are simplified by those of TCC and AOD in these simulations. The spatial patterns of the aerosol contributions simulated by the radiative transfer model match much better with the CERES fields than those of cloud contributions, although some discrepancies exist in their magnitudes (Figure 5a,b,e,f). Reasons for these differences include different AOD input datasets (CERES-derived SSR is retrieved from MATCH AOD at 550 nm and 840 nm,

but the model-estimated SSR is calculated from MODIS AOD only at 550 nm), distinct model assumptions (e.g., the inability to allow the AOD vertical-weighted profiles to vary annually), and the model's inabilities in considering the interactions between clouds and aerosols, etc. However, the simulated TCC contributions to SSR trends is well performed both in their distributions and magnitudes in most regions of northern and northeastern China, but with large discrepancies over southern China when compared to those of CERES (Figure 5a,e). This is likely because cloud cover is not the only variable to induce cloud effects on SSR trends, other cloud properties, like liquid/ice effective radius, cloud optical depths, liquid/ice water content, etc., also contribute to it, as well as the poor performance in the practical vertical overlaps of clouds, and the inability to consider the interactions of clouds and aerosols in the model.

To summarize the above, the radiative transfer model can generally simulate the aerosol effects on SSR trends during 2005–2018 over the eastern half of China using the annual mean inputs of AOD. The simulated effects of cloud cover from all cloud types agree reasonably well with the SSR trends induced by the clouds in most regions of northern and northeastern China, except for southern China. This can be well explained by the reasons mentioned above. Thus, further study is still needed to explore the cloud effects on SSR trends.

*3.4. The Relative Contributions of Cloud Cover from Different Cloud Types to the Annual and Seasonal ACC-Induced SSR Trends by a Radiative Transfer Model*

This study aims to explore the relative contributions of cloud cover from different cloud types to the ACC-induced SSR trends using the radiative transfer model. Figure 6 illustrates the simulated annual SSR trend percentages caused by HCC, mid-HCC, mid-LCC, and LCC relative to the ACC-induced SSR trends and the SSR trend percentages due to ACC relative to the total SSR trends during 2005–2018 in China. The SSR increases caused by the declines in HCC (Figure S7a) and especially LCC (Figure S7d) are the primary contributors to the simulated ACC-induced SSR increases over eastern, northern, and northeastern China during this period (Figure 6a,d,e), while contributions from mid-HCC and mid-LCC play opposite roles except for some individual regions (Figure 6b,c and Figure S7b,c). However, the simulated decreases in ACC-induced SSR over southern China (Figure 6e) are mainly from the contributions of HCC, mid-HCC, and particularly mid-LCC (Figure 6a–c and Figure S7a–c), whereas contributions from LCC have an opposite effect (Figure 6d and Figure S7d).

The contributions of cloud cover from different cloud types to the simulated ACC-induced SSR trends mainly depend on sub-regions and their seasonal variations. Similarly, as can be seen from supplementary Figures S8–S11 for the seasonal contributions, the variation of LCC is undoubtedly the major cause of the simulated ACC-induced SSR changes all year round over the eastern half of China, especially in winter. Besides central and eastern China in the spring season, changes in HCC overall have positive effects on the simulated ACC-induced SSR variations. Interestingly, the mid-LCC variations generally show contrary impacts with that of LCC in addition to the spring season. However, the contributions from mid-HCC over studying regions seem much more irregular compared to the other cloud types. This is likely due to the complicated heating and cooling roles of the mixed-phase clouds in the atmosphere.

To determine the relative contributions of cloud cover from different cloud types to the ACC-induced SSR trends averaged over some specific regions in detail, four typical sub-regions showing large discrepancies among different cloud types are selected in this study with symbols of 1–4 as shown in Figure 6, which represent the North China Plain, East China, southwestern China, and South China, respectively. The simulated annual and seasonal relative contributions of cloud cover from different cloud types averaged over these four sub-regions are shown in Figure 7. As can be seen from Figure 7a, the simulated annual and seasonal ACC-induced increasing SSR trends over the North China Plain (Figure 6e, Figures S8–S11e) are largely due to the reductions in LCC, and the SSR trend contributions are generally larger than 60%, with the largest contribution even reaching

74% in summer. Overall, the SSR trends due to LCC and HCC show strong and slightly positive contributions, respectively, while mid-LCC appears to provide slightly negative contributions both on annual and seasonal time scales. However, contributions from mid-HCC vary from season to season, with slightly positive contributions in summer and winter as well as negative effects for the other seasons, with up to 32% for autumn. The distributions of annual SSR contributions due to different cloud types are similar to those in spring and autumn, although mid-HCC contributing more in autumn, whereas the distributions are alike in summer and winter, despite slightly larger contributions from HCC and mid-HCC in winter.

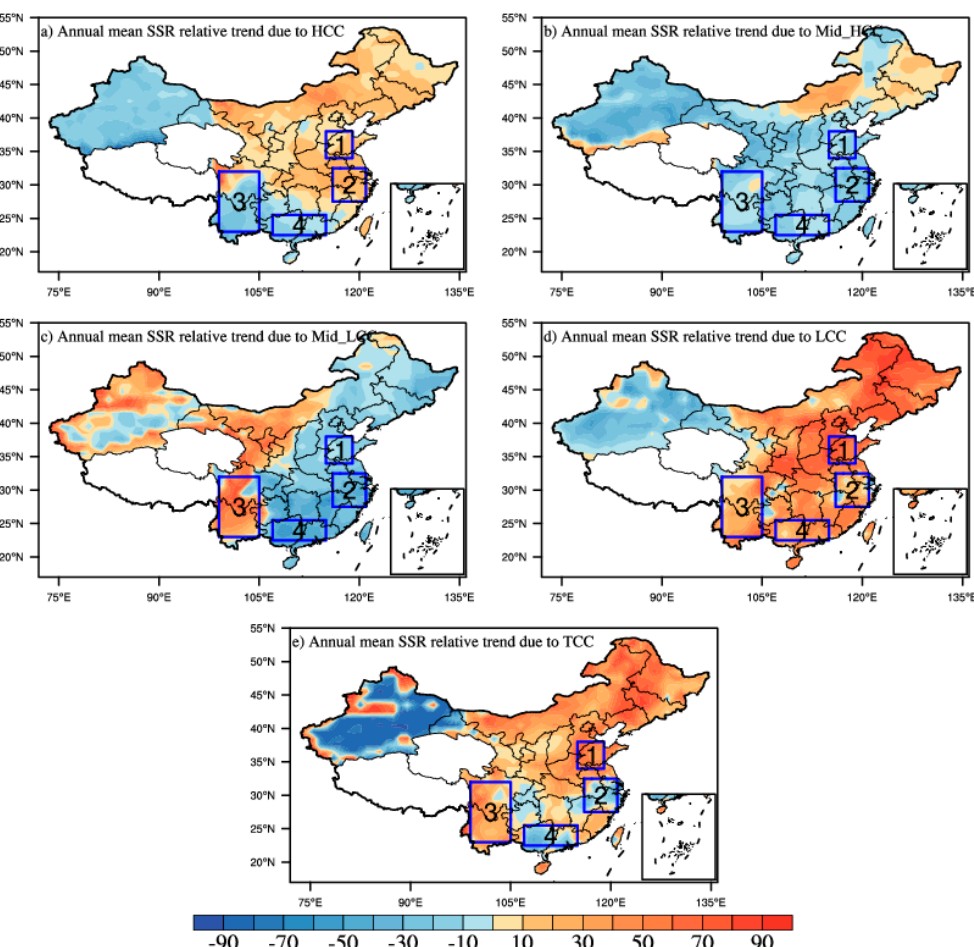

**Figure 6.** Annual mean simulated SSR trend percentages (unit: %) due to (**a**) high cloud cover (HCC), (**b**) mid-HCC, (**c**) mid-low cloud cover (LCC), and (**d**) LCC relative to the sum of absolute trends due to each type of cloud cover (ACC-induced SSR trends), as well as simulated SSR trend percentage (unit: %) due to (**e**) cloud cover from all cloud types relative to the sum of absolute trends due to each driving factor (total SSR trends) from 2005 to 2018 over China. The blue labeled sub-regions represent the (1) North China Plain, (2) East China, (3) southwestern China, and (4) South China, which occupy latitudinal ranges of 34–38°N, 27.5–32.5°N, 23–32°N, and 22.5–25.5°N, and longitudinal ranges of 115–119°E, 116–121°E, 99–105°E, and 107–115°E, respectively.

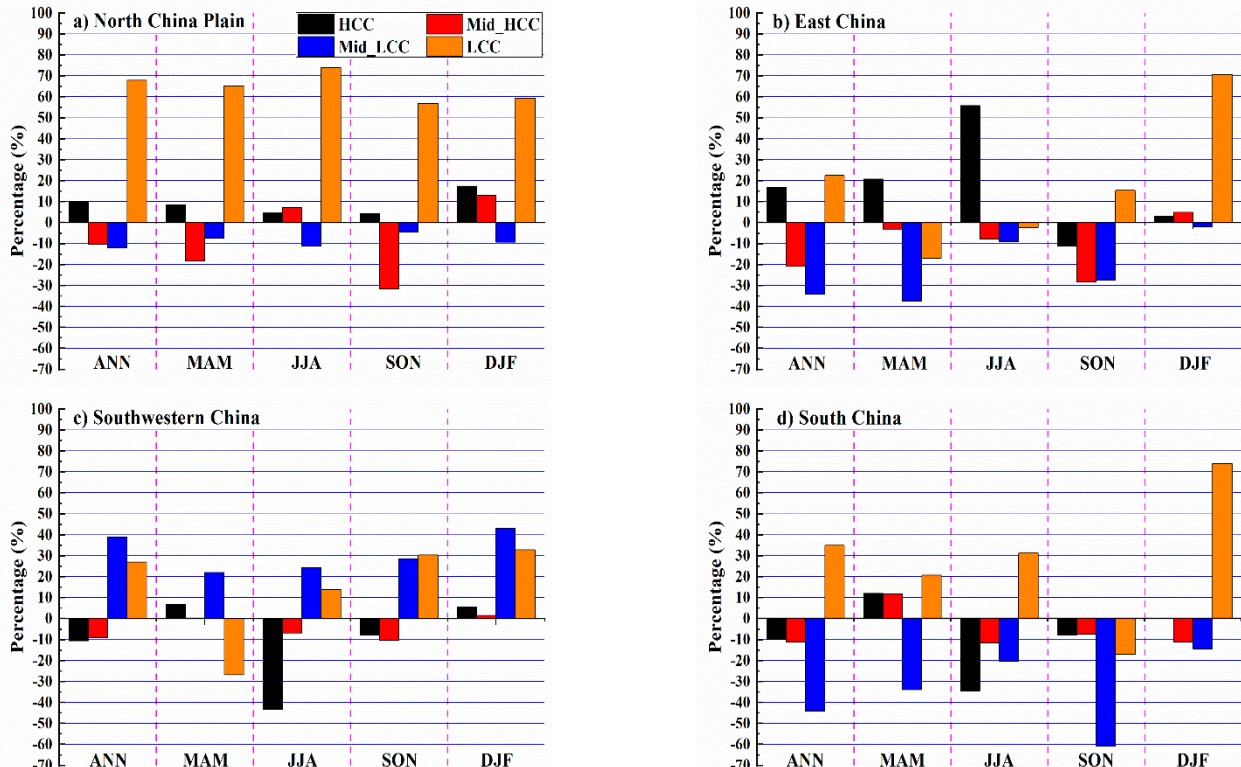

**Figure 7.** Annual (ANN) and seasonal (MAM, JJA, SON, DJF) simulated SSR trend contributions (unit: %) from HCC (black), mid-HCC (red), mid-LCC (blue), and LCC (orange) relative to the sum of absolute trends due to each cloud cover type (ACC-induced SSR trends) averaged over four sub-regions for 2005–2018 in China, respectively, including (**a**) North China Plain, (**b**) East China, (**c**) Southwestern China, and (**d**) South China.

For East China, the contributions of cloud cover from different cloud types greatly vary from each other at annual and seasonal time scales (Figure 7b). The annual contributions of HCC and LCC are positive, while other cloud types show negative contributions with percentages of no more than 35%. The negative contribution of mid-LCC is the main cause (37%) of the simulated ACC-induced declines in SSR trends in spring, followed by LCC and mid-HCC (very small), and the contribution of HCC is positive with a similar contribution percentage as that of LCC. In summer, the decreases in HCC account for 56% of the simulated ACC-induced SSR trends, while contributions from other cloud types appear to have very small negative impacts. The negative contributions from mid-HCC and mid-LCC contribute a little more to the SSR trends in autumn when compared to the only slightly positive contribution of LCC. In winter, the variations in LCC dominate the simulated increasing SSR trends with a percentage reaching 70%, while contributions of cloud cover from other cloud types can be neglected.

For southwestern China (Figure 7c), annual and autumn mean contributions are alike, both reductions in mid-LCC and LCC show much larger positive contributions than negative contributions from HCC and mid-HCC. The SSR trends in spring are attributed to the positive and negative contributions of mid-LCC and LCC, respectively, followed by a slightly positive contribution from HCC. A significantly negative HCC contribution occurs in summer, followed by much smaller positive contributions from mid-LCC and LCC. All cloud types show positive contributions to ACC-induced SSR trends in winter, especially for mid-LCC and LCC being up to 44% and 33%, respectively.

The contributions over South China are featured by negative contributions of mid-LCC at both annual and seasonal time scales and positive ones of LCC except for autumn (Figure 7d). The annual negative contribution of mid-LCC is greater than the positive contribution of LCC. Thus, the annual SSR over this region tends to decrease combined

with the much smaller negative contributions from HCC and mid-HCC. In spring, mid-LCC shows a much larger negative contribution than the positive contributions from LCC, HCC, and mid-HCC. In summer, only LCC exhibits a positive contribution and is smaller than the negative contribution of HCC. All cloud types in autumn appear to have negative contributions, with the largest contribution from mid-LCC reaching 60%. A marked positive contribution of LCC accounts for 74% of the ACC-induced SSR trends in winter, while other cloud types have much smaller negative contributions.

In general, from the perspective of different time scales and sub-regions, the decreases in LCC have the largest positive contributions to the ACC-induced SSR trends over the North China Plain irrespective of the annual or seasonal time scales. For East China, the ACC-induced SSR increases are mainly from positive contributions of HCC in summer and LCC in winter, respectively. The positive contributions from mid-LCC and LCC are attributable to the annual, autumn, and winter mean ACC-induced SSR trends over southwestern China, while increases in HCC have a marked negative contribution in summer. The increases in mid-LCC play substantial roles in the simulated annual, spring, and autumn mean ACC-induced SSR trends over South China, whereas the decline in LCC is the major contributor to that in winter.

## 4. Conclusions and Discussions

The annual SSR variations under all-sky, clear-sky, all-sky-no-aerosol, and clear-sky-no-aerosol conditions, as well as their associated driving factors (e.g., cloud cover from different cloud types, AOD, water vapor, and $O_3$) derived from satellite products during 2005–2018 over China, are investigated to identify the major causes of the SSR trends. Then, the annual and seasonal means of the above driving factors for 14 years are used as inputs into a radiative transfer model to calculate their respective effects on the SSR trends by using a concept of relative SSR trend percentage to avoid discrepancies caused by different model assumptions. Furthermore, this study examines the relative contributions of cloud cover from different cloud types to the simulated ACC-induced SSR trends averaged over four typical regions of China both at annual and seasonal time scales.

Compared to the ground-based observations from the CMA, the CERES SYN1deg product can generally be regarded as credible during this period over the eastern half of China. The satellite-derived results confirm the primary roles of clouds and aerosols in SSR trends during this period, while those of water vapor and $O_3$ are much smaller in their magnitudes. A nationwide brightening appears over most regions in China and the clouds and aerosols are the major causes of this increase over northern and northeastern China, albeit stronger aerosol effects can be seen in individual regions. However, the cloud effects contribute more to the all-sky SSR reductions over southern China.

The radiative transfer model can generally simulate the aerosol effects on SSR trends from the CERES SYN1deg product during 2005–2018 over the eastern half of China by the annual mean inputs of AOD, while the inputs of annual mean all types of cloud cover can reproduce the SSR trends induced by the clouds in most regions of northern and northeastern China, in addition to southern China. The model assumptions, different input data sources, and inadequacies in the assumptions with respect to the vertical cloud overlaps, as well as the interactions between clouds and aerosols in the model, are the likely causes for the poor simulations of clouds over southern China.

The relative contributions of cloud cover from different cloud types to ACC-induced SSR trends simulated with the single-column radiative transfer model vary among different regions in China. The significant positive contribution of LCC over the North China Plain is the major contributor to the simulated annual and seasonal ACC-induced increases in SSR trends, with percentages generally greater than 60%. For East China, the marked ACC-induced SSR increases in summer and winter are primarily due to the strong positive contributions from HCC and LCC, with percentages reaching 56% and 70%, respectively, while the declines in SSR during other seasons result from the combined effects of all cloud types, especially increases in mid-LCC and mid-HCC. The magnitudes of the relative

contributions from all cloud types averaged over southwestern China seem to be much more uniform. The increases in HCC in summer are attributable to the simulated ACC-induced decreasing SSR trends, whereas the changes in mid-LCC and LCC are responsible for the simulated increasing SSR trends for other seasons. The declines in LCC and increases in mid-LCC dominate the simulated ACC-induced increases and declines in SSR trends averaged over South China in winter and autumn with percentages of 74% and 60%, respectively. The negative mid-LCC contributions except for HCC in summer and positive LCC contributions together result in the ACC-induced SSR declines over South China in other seasons.

Overall, the simulated relative contributions of cloud cover from different cloud types to the ACC-induced SSR trends over China in recent years largely depend on sub-regions and seasons, and the changes in HCC usually contribute more to the ACC-induced SSR trends in summer over most regions in China, which is possibly associated with the deep convection in this season. However, the contributions from LCC or mid-LCC are responsible for the ACC-induced SSR trends over most regions all year round.

The vast majority of the climate models tend to overestimate the SSR from surface direct observations, which has been a long-standing problem over several decades [64]. Hopefully, this study would provide some implications for a regional or global climate model with a radiation module to reduce the bias of the solar radiation reaching the surface. The possible reasons for the discrepancies between the model-estimated and satellite-derived SSR might also help improve the accuracy of surface energy budgets in the climate systems, thereby contributing to a more credible climate model.

**Supplementary Materials:** The following are available online at https://www.mdpi.com/2072-4292/13/4/704/s1, Figure S1–S4: Validation of seasonal average surface solar radiation (SSR) and their corresponding trends under all-sky conditions from CERES satellite-derived product using ground-based observations, Figure S5: Map of provinces in China, Figure S6: Annual mean distributions of different driving factors from various satellite-derived products, Figure S7: Annual mean trends of different types of cloud cover from CERES-derived product, Figure S8–S11: Same as Figure 6, but for seasonal means.

**Author Contributions:** Wrote the manuscript, Q.W.; proposed the ideas and gave some comments and suggestions to the manuscript, Q.W., H.Z., and M.W.; provided the ground-based all-sky SSR data and the simulated AOD vertical distributions for the year 2006, respectively, S.Y. and Y.C.; contributed to the better understanding of the radiative transfer model, Q.C.; X.Z. and G.S. All authors have read and agreed to the published version of the manuscript.

**Funding:** This research was funded by the National Key Research and Development Program of China (Grant Number: 2017YFA0603502), the (Key) National Natural Science Foundation of China (Grant Number: 91644211), and Science and Technology Development Fund of Chinese Academy of Meteorological Sciences (Grant Number: 2021KJ004). Global dimming and brightening research at ETH Zurich obtaining fundings from a sequence of Swiss National Science Foundation Grants (Grant No 200021_135395, 200020_159938, 200020_188601) and from the Federal Office of Meteorology and Climatology MeteoSwiss within the framework of GCOS Switzerland.

**Data Availability Statement:** The CERES SYN1deg data is available at https://ceres-tool.larc.nasa.gov/ord-tool/jsp/SYN1degEd41Selection.jsp (accessed on 1 January 2021); The AIRS data is accessible from https://disc.gsfc.nasa.gov/datasets/AIRS3STM_006/summary?keywords=AIRS (accessed on 1 January 2021); The MODIS data is from https://ladsweb.modaps.eosdis.nasa.gov/archive/allData/61/MYD08_M3/?process=ftpAsHttp&path=allData%2f61%2fMYD08_M3 (accessed on 1 January 2021); The CloudSat data is from http://www.cloudsat.cira.colostate.edu/data-products/level-2b/2b-cwc-ro (accessed on 1 January 2021); The MERRA2 dataset is obtained at https://disc.gsfc.nasa.gov/datasets/M2IMNPANA_5.12.4/summary?keywords=merra-2 (accessed on 1 January 2021). The ERA-Interim is from https://apps.ecmwf.int/datasets/data/interim-full-moda/levtype=sfc (accessed on 1 January 2021).

**Conflicts of Interest:** The authors declare no conflict of interest.

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
