# Peer review of "Potential Driving Factors on Surface Solar Radiation Trends over China in Recent Years"

_remotesensing, doi:10.3390/rs13040704_

Round 1
Reviewer 1 Report
Please find my comments in the attached file.

Reviewer 2 Report
The manuscript by Wang et al. analyses mean surface solar radiation (SSR) over China during 2005-2018 together with auxiliary data from various data sources including satellite and in situ measurements together with modelled data. It mainly focuses on the effects of clouds and aerosols on the changes in SSR over time. The second half of the paper concentrates on the effect of different cloud types on SSR. The investigation of changes in SSR and its causes is important and scientifically relevant.
The aim and novelty of the work should be clearly stated by the authors. Even though the authors declare the novelty of analysis of different cloud types, this covers only one part of the paper, the rest is a confirmation on a long list of previous works, detailed in the Introduction section.
The major issue with the manuscript is the lack of addressing the accuracy/uncertainty of the data used as well as the performance of the used models. There are very many different data sources used, and very little information about them and their uncertainty. Even though for example in the case of CMA in situ data there are references to two papers concerning data quality control and homogenization, some basic information should be included in this paper. As the presented trends are very small and mainly statistically insignificant, the authors should address the uncertainty of the data. For SYN1deg, biases of 5.7% and 2.9% compared to ground measurements depending on the wavelength have been reported, so are the presented trends significant in that perspective?
In section 3.1 validation of CERES dataset against in situ data is presented, but no sophisticated quantification of the validation is given, beside the visual on Figure 1 and single numbers in Table 2. The same problem is with the comparison of BCC_RAD radiative transfer model results with values from CERES products (Section 3.3.). There are annual mean trend percentages given in Table 2 for the selected four regions, but more comprehensive analysis is needed. There are areas with much larger differences (even opposite trends) than the selected regions. In the case of the model input data (clouds, aerosols gases etc.) no information about uncertainty has been given. The largest issue would be with cloud cover on four different levels – how well the data represents the reality and if any conclusions based on these analyses could be drawn. The concerning issue is with clouds is highlighted in Figure 3Sa and b, where the trend of high cloud fraction turns sign along a straight line dividing west and east part of China. In addition, the validation of a model should be against ground measurements or at least both, ground measurements and satellite data.
The final general comment is about the figures. As the paper is about trends, it would benefit from a time series figure. For example, a line graph with the annual area-averaged SSR with standard deviation from 2005 to 2018 to show the fluctuation of the values and give a better overview of the reported results.
Specific comments:
P1, L19-20: “possible causes” and “likely drivers” are the same thing in one sentence. I would suggest “… clear-sky-no-aerosol conditions are analysed during……”
P2, L52: I suggest changing “mass” to “mass concentration” or adding a separate phrase to emphasize the role of changing aerosol loading in the atmosphere if the mass indicates that the average mass of a particle has changed.
P2, L53-55: Rephrase the last sentence to showcase the difference from your work. Right now, it seems you contradict these articles, and I don’t see this as the summary of these papers.
P2, L83-84: “More common studies tend to suggest that the brightening trends in SSR over China are mainly due to the combined effects of clouds and aerosols.”. The same is true for the studies mentioned in the previous paragraph, so the division here is not clear.
P3, L105: “..2005 to 2018”. As the start and end are great importance for trend analysis, the reasoning behind the choice of the period should be addressed.
P3, L105: “…yearly annual and seasonal means…”. The phrase is hard to read here and throughout the manuscript, “yearly” should be dropped and the sentence could be rephrased to show that the values were calculated for each year.
P3, L142: basic details about the accuracy of in situ measurements should be added here.
P4, L193: “..yearly annual..”. Same comment as P3, L105
P6, Figure 1: I would consider changing the colour scheme for Figure 1a. The red and blue shades are good for Figure 1b for positive and negative values, but distracting in Fig. 1a.
P6, L254-256: “..annual mean values of the all-sky SSR from the CMA stations are generally close to those of the CERES-derived averages, except for some individual stations over the northwestern regions.”. Even though Figure 1 is nice, this needs quantification, maybe a distribution (histogram, scatterplot etc.) figure as there are 99 stations.
P6, L257: “…very well…”. Again, this is not enough to describe the goodness of a dataset.
P6, L262: “..agree well…”. Again, this is not enough to describe the goodness of a dataset. Here, even visually it can be seen, that the fit is not as good as for the annual average.
P7, L271-276: For comparison, it would be maybe more accurate to describe the difference at ground-based measurement points. Due to the large study area with very different conditions in different parts of it, the area-weighted average trend is maybe not the best characteristic, especially as the trend is in some cases positive and some cases negative.
P7, L286-289: There is no analysis showing the different behaviour of CERES data over western and eastern China compared to ground-based data to draw the conclusion: “….generally they correspond well to the ground-based observations both at the annual and seasonal time scales during this period over the eastern half of China,…” and “…CERES product over the eastern half of China can be used as reference data….”. Either regional analysis or reasoning for the selection of the eastern part should be added.
P8, Figure 2: Here, the entire region is included, although previously it was stated, that only the eastern part is good for further analysis.
P8, Figure 3: For proxy data, for better understanding of the differences in SSR, it would be useful to add figures for average values, in addition to trends.
P10, L371: “China-mean contribution” – rephrase
P10, L391: “TP”, unintroduced abbreviation. It is unclear, why this area is masked out, as in Fig. 2 it shows a statistically significant trend in each condition.
P11, L413: It is unclear what the unitless CERES SYN1deg product is.
P11, L420-421: “…the effects of clouds and aerosols on SSR trends are simplified by those of TCC and AOD in these simulations.” How?
P12, L458: Section 3.4. I would consider rewriting this section as it is hard to follow and the explanations are repeating. I would recommend shortening and generalizing the description on P13 L481 – P14 L512 and keeping the detailed analysis for the regions.
P12, L464-465: “The SSR increases caused by the declines in HCC (Figure S3a)….”. What is the cause for the sharp (seemingly unnatural straight-line) difference in Figure S3a and partly S3b between the eastern and western part of China? Without knowing the cause, the results give doubt to the goodness of the model and to the value of the analysis in Section 3.4.
P13, Figure 5: Why are the selected regions different from the previous analysis (Figure 4)? For the consistency, it would make sense, in my opinion, to choose one set and keep with it. Choosing only small areas for analysis to give a general assessment for the entire region is a difficult task as it is and the change is not explained. It must be kept in mind, that the results would be quite different in other areas.
P17, L632-634. The goal was not achieved. Even though the manuscript shows the results from the different sources, it lacked the effort to estimate the accuracy and therefore the relevance of the results.
Reviewer 3 Report
Dear Authors,
I found the presented manuscript to be very well prepared and structured. Especially the introduction part gives a thorough and easy to follow overview of the current state of the field while the conclusions give a good summary of the study.
I found some minor issues that should be resolved before publication.
Specific comments:
line 126: how do You define 'major' greenhouse gases?; why are CFCs excluded from that group while their global contribution to global warming is greater than that of O3 and also probably then N2O?; also the commas separating the chemical formulae in the brackets look inverted on my computer for some reason.
line 145: the word 'beware' is probably a colloquialism here; please consider 'be advised' or 'notice'
line 156: as the rest of the sentence is kept in the singular form please change 'numbers' to 'number'
line 175: 'above 100 (1) mb' - what does the (1) stand for?
line 200: 'while it is mainly' - what does 'it' refer to here?
line 204: 'AOD vertical weight profiles' - what is an AOD 'weight' profile?; I am not familiar with the term 'AOD weight'
line 226: '??â„Ž' - please change to 'ith value'
line 344: 'please change 'cloud' to 'could'
line 363: a comma is missing in 'mean median'
line 523: please change 'strongly' to 'strong'
